# Effect of Antimicrobial Use in Conventional Versus Natural Cattle Feedlots on the Microbiome and Resistome

**DOI:** 10.3390/microorganisms11122982

**Published:** 2023-12-14

**Authors:** Catrione Lee, Rahat Zaheer, Krysty Munns, Devin B. Holman, Gary Van Domselaar, Athanasios Zovoilis, Tim A. McAllister

**Affiliations:** 1Lethbridge Research and Development Centre, Agriculture and Agri-Food Canada, Government of Canada, 5403 1st Avenue South, Lethbridge, AB T1J 4B1, Canada; catrione.lee@agr.gc.ca (C.L.); rahat.zaheer@agr.gc.ca (R.Z.); krysty.thomas@agr.gc.ca (K.M.); 2Southern Alberta Genomic Sciences Centre, Department of Chemistry and Biochemistry, University of Lethbridge, 4401 University Drive West, Lethbridge, AB T1K 3M4, Canada; athanasios.zovoilis@uleth.ca; 3Lacombe Research and Development Centre, Agriculture and Agri-Food Canada, Government of Canada, 6000 C and E Trail, Lacombe, AB T4L 1W1, Canada; devin.holman@agr.gc.ca; 4National Microbiology Laboratory, Public Health Agency of Canada, Government of Canada, 1015 Arlington Street, Winnipeg, MB R3E 3R2, Canada; gary.vandomselaar@phac-aspc.gc.ca

**Keywords:** antimicrobial resistance, livestock production, microbiota, resistome, mobilome, metagenomic sequencing, raised without antibiotics

## Abstract

Antimicrobial use (AMU) in the livestock industry has been associated with increased levels of antimicrobial resistance. Recently, there has been an increase in the number of “natural” feedlots in the beef cattle sector that raise cattle without antibiotics. Shotgun metagenomics was employed to characterize the impact of AMU in feedlot cattle on the microbiome, resistome, and mobilome. Sequenced fecal samples identified a decline (q < 0.01) in the genera *Methanobrevibacter* and *Treponema* in the microbiome of naturally vs. conventionally raised feedlot cattle, but this difference was not (q > 0.05) observed in catch basin samples. No differences (q > 0.05) were found in the class-level resistome between feedlot practices. In fecal samples, decreases from conventional to natural (q < 0.05) were noted in reads for the antimicrobial-resistant genes (ARGs) *mef*A, *tet*40, *tet*O, *tet*Q, and *tet*W. Plasmid-associated ARGs were more common in feces from conventional than natural feedlot cattle. Interestingly, more chromosomal- than plasmid-associated macrolide resistance genes were observed in both natural and conventional feedlots, suggesting that they were more stably conserved than the predominately plasmid-associated tetracycline resistance genes. This study suggests that generationally selected resistomes through decades of AMU persist even after AMU ceases in natural production systems.

## 1. Introduction

Antimicrobial resistance (AMR) is a global health crisis that impacts clinical and animal health, food security, and the environment. By the year 2050, it is expected that antimicrobial-resistant infections will result in 10 million deaths per year, overtaking cardiovascular disease and cancer as the leading cause of death [1]. Antimicrobial resistance surveillance programs and risk assessments have been launched in a number of countries, including Canada [2], the United States [3], the European Union [4], and at a global scale [5,6,7].

Identification of antimicrobial resistance genes (ARGs) using metagenomics is one approach to characterizing the nature of AMR across the One Health Continuum. Metagenomics can be used to compare information on phylogeny, ARGs, mobile genetic elements (MGE), and virulence factors across environments. If sequencing depth and read length are sufficient, the context of ARGs can also be established, providing information on horizontal gene transfer. Of these, ARGs are most frequently transferred via conjugation through the action of a number of MGEs, including plasmids, integrative conjugative elements (ICE), transposons, integrons, and insertion sequences [8].

There is a concern with regard to the contribution of AMU in livestock to overall AMR [9]. In beef cattle, the majority of AMU takes place in feedlots as opposed to the more extensive cow–calf sector [10,11]. While feedlots can house over 40,000 head of cattle, which allows for higher meat production per head [12], these high densities subject individuals to stressors that can encourage bacterial pathogens to proliferate and rapidly transfer within a herd [13]. The resulting increased incidence of morbidity requires much higher AMU to combat infections through therapeutic, prophylactic, and metaphylactic practices as compared to the more extensively managed cow–calf sector [14]. As the majority of antimicrobials in beef cattle production are administered to feedlot cattle, it is a logical point to investigate AMR within this portion of the beef production cycle [15]. However, given the current knowledge about how AMU increases the prevalence of ARGs, some feedlots have opted to employ natural management practices that prohibit the use of antimicrobials. However, the effectiveness of this practice in limiting the prevalence of ARGs in feedlots is inconclusive [16,17,18].

Shotgun metagenomics is one of the most comprehensive methods to study microbiomes, resistomes, and associated mobilomes [19]. Sampling all of the DNA from a specific environment considers the contribution of the genomes of uncultured bacteria to the resistome, generating a perspective of all known ARGs within a sample or environment. In addition, the degree of AMU may also impact the colocalization of ARGs and MGEs. Some studies have aimed to compare the effect of no AMU in livestock production to the conventional use of antimicrobials on the resistome but found no impact on meat trimmings or fecal resistomes [17,20].

The goals of this study were to investigate the effect that beef cattle raised without antimicrobials have on the microbiome, resistomes, and associated mobilome of beef feedlot-associated environments. We accomplished this by directly comparing fecal composite and catch basin water samples from conventional (raised with antimicrobials) and natural (raised without antimicrobials) beef cattle feedlots in Alberta, Canada. We predicted that AMU in conventional beef feedlots would result in increased prevalence and abundance of ARGs and ARGs colocalized with MGEs as compared to natural feedlots.

## 2. Materials and Methods

### 2.1. Natural and Conventional Feedlots

Conventional (CONV; *n* = 2) and natural (NAT; *n* = 2) feedlots in Alberta were included in this study. Cattle in one NAT feedlot were fed a typical North American backgrounding diet consisting of 62% corn silage, 27% barley grain, 7% vitamin mix, and 4% canola meal and received no antimicrobials, ionophores, or hormonal implants. Steers or heifers that required antimicrobials due to clinical illness were removed from the general population and quarantined in a hospital pen until harvest. Both CONV and one NAT feedlot cattle were fed a finishing diet containing 85% barley grain, 10% barley silage, and 5% supplement. The CONV diet also contained the ionophore monensin at 48 ppm and the macrolide tylosin at 11 ppm (Elanco Animal Health, Greenfield, IN, USA). For heifers, the supplement also contained 0.045 ppm melengestrol acetate (MGA). For the last 40 days of the feeding period, ractopamine hydrochloride was included in the diet at 30 ppm. Cattle also received an implant containing 200 mg of trenbolone acetate and 20 mg of estradiol (REVALOR^®^-200, Merck & Co., Inc., Rahway, NJ, USA). An average of 180 head of cattle were housed in open pens and all cattle had free access to water and feed. Pen-level metadata (sex and age average) for cattle used to collect fecal composites can be found in Appendix A.

### 2.2. Fecal Collection for Conventional and Natural Feedlots

Composite fecal samples (20 g) were collected from pens (*n* = 10) in both CONV (*n* = 2) and NAT (*n* = 2) feedlots over two years (August 2016–July 2018; CONV *n* = 30 and NAT *n* = 30). Each 20 g composite was generated by thoroughly mixing 1 g samples from 20 randomly selected fresh fecal pats within each pen. Composite fecal samples were placed in sterilized Whirl-Pak^®^ bags (Sigma-Aldrich, St. Louis, MO, USA) (532 mL) and transported on ice to the laboratory within 4 h of collection. Upon arrival, feces were homogenized, flash frozen in liquid nitrogen, and stored at −80 °C in flat sheets.

### 2.3. Metagenomic DNA Extraction and Sequencing

To isolate metagenomic DNA from homogenized and frozen composite fecal samples, 325 mg of sample was transferred into a 2.0 mL sterilized safe-lock snap-cap with 0.4 g of sterilized zirconia beads (0.3 g of 0.1 mm and 0.1 of 0.5 mm sizes). Metagenomic DNA extraction and PCR inhibitor removal were performed following a previously described procedure [21]. The quantity of the extracted DNA was determined by fluorescence at 480 nm using Quant-iT PicoGreen fluorometer (Thermo Fisher Scientific, Mississauga, ON, Canada), and the quality/purity was determined, using a NanoDrop spectrophotometer (Thermo Fisher Scientific), by measuring the ratios of absorbance at 260/280 nm and 260/230 nm wavelengths. Extracted DNA samples with absorbance ratios at 260/280 nm and 260/230 nm of 1.7–2.0 and 2.0–2.2, respectively, were considered acceptable. To evaluate the presence of PCR inhibitors undiluted and various dilutions of the extracted DNA were used as PCR templates to amplify 16S rRNA genes with the universal bacterial primers 27F and 1492R [22].

Genome Québec Innovation Centre (Montréal, QC, Canada) performed all library preparations, sequencing, and quality control steps. Metagenomic sequencing libraries were prepared using a PCR-free shotgun DNA library preparation kit (Lucigen, LGC Biosearch Technologies; distributors: VWR International, Radnor, PA, USA). Prepared libraries were sequenced on an Illumina NovaSeq 6000 platform, with 45 samples multiplexed per sequencing lane to generate 2 × 150 base paired-end (PE) sequence reads. Each sequencing lane was spiked with the PhiX174 *sensu lato* viral genomic DNA library at ~1% concentration of the total DNA loaded per lane for the quality control of cluster generation and sequencing. Sequencing read data are available in the National Center for Biotechnology Information (NCBI) Short Read Archive (SRA) under the BioProject ID PRJNA420682.

### 2.4. Bioinformatics Resources

Metagenomic sequence information was stored in the Integrated Rapid Infectious Disease Analysis (IRIDA) Platform [23] at the National Microbiology Laboratory’s (NML; Public Health Agency of Canada) high-throughput computing cluster. Microbiome taxonomic and resistome profiling was carried out using the workflow outlined below.

Trimmomatic v0.36 [24] was used to trim adapters from PE reads and filter out low-quality reads using the following parameters: leading and trailing adapters with “N” bases or a quality score < 3 were trimmed from sequence reads; a sliding window quality score filtered every 4 bases with a minimum Phred score of 15; sequences with <36 nucleotides were discarded; adapters supplied in the TruSeq3 adapter sequence file with a maximum of 2 mismatches in the initial seed were removed; and if a match score of 30 was reached, the adapter was clipped. Singletons, whereby a read’s matching pair failed quality control, were also included in downstream analysis.

In order to remove the Illumina PhiX spike-in control, reads were filtered against the *Escherichia* phage PhiX174 genome (GenBank accession NC_001422.1) using the minimum exact match (MEM) algorithm of the Burrows-Wheeler aligner (BWA) v0.7.17.1 [25]. The sorted alignments were then processed with SAMtools v2.0.2 [26] to retain only those reads that did not map to the PhiX174 genome. This was carried out using a flag value of 4 to extract the unmapped reads in binary alignment map (BAM) format. The PE reads that did not map to PhiX174 were then extracted from the alignment using the bamToFastq tool within BEDTools v2.27.0.0 [27]. The PhiX-filtered reads were then classified with Kraken v2.0.8-beta [28] using the custom Kraken database bvfpa [21]. Kraken2 results were then filtered using a confidence threshold of 0.05 to select for taxonomic assignments http://ccb.jhu.edu/software/kraken/MANUAL.html (accessed on 19 August 2021) [21]. All Chordata reads were filtered out in downstream analyses.

Resistome analysis was conducted in parallel with the taxonomic classification as follows: trimmed PE reads were mapped to ARG sequences in the MEGARes database v2.00 [29] using BWA-MEM v0.7.17.1 [25]. The alignments in BAM format were converted to a sequence alignment map (SAM) v2.0.2 format and post processed with the Coverage Sampler tool (https://github.com/cdeanj/coveragesampler; accessed on 19 August 2021) using a 75% gene fraction threshold [30]. These output matrices of the resistome and taxonomic composition were then analyzed on a local installation of R (v4.3.1; http://www.r-project.org/; accessed on 19 October 2023).

### 2.5. Assembly of Contigs

For metagenomic assemblies, adapters were trimmed from PE reads using fastp v0.20.1 with the following parameters: leading and trailing adapters were trimmed when “N” bases or quality scores were <15; a sliding window quality score was filtered for every four bases with a minimum Phred score of 15; and sequences with <100 nucleotides were discarded. Host and PhiX174 reads were filtered out using the *Bos taurus* (NC_037328.1) and PhiX (NC_001422.1) genomes concatenated with Bowtie2 v2.3.4.3 [31], when minimum and maximum fragment lengths for valid PE alignments were 0 and 500, respectively. SAMtools flags ‘-f 12 -F 256’ were used to convert SAM files into BAM (binary alignment map) files and only unmapped reads were included for downstream analyses. SAMtools was then used to sort the sequences in the BAM files to enable BEDTools to output PE FASTQ-formatted sequence files. Contigs were assembled with the MEGAHIT v1.2.9 assembler [32] with a minimum kmer length of 3 bp and minimum contig length of 1000 bp.

### 2.6. Microbiome and Resistome Analysis

Taxon abundances were normalized using the trimmed mean of M-value (TMM) method [33] as per Pereira et al. [34]. For visualizations, phyla present at <1% relative abundance were aggregated together into an ‘Other’ group. Species-level classifications were included in the diversity analyses. Alpha diversity was measured with the Shannon diversity index and beta diversity was measured using the Bray–Curtis dissimilarities calculated in R v4.3.1 with vegan in the phyloseq package v1.44.0. Log2 fold change was computed by the ANCOMBC v2.2.2 R package treating CONV feedlots as the reference. Fold change was calculated and reported in tables. For decreases in NAT feedlots (i.e., fold change < 1) the inverse was taken to ease in text readability.

From the AMR++ v0.1 workflow, the tool MEGARes v2.0 [29] was aligned against a database containing ARGs to antibiotics, biocides, metals, and multi-compound resistance (an ARG conferring resistance to a combination of antibiotics, biocides, and metals). This database contained the following descending ARG classifications: type, class, mechanism, and group. Any ARGs based on single nucleotide polymorphisms (SNPs) were removed. Additionally, any ARGs that contained the word “regulator” were removed from the dataset due to the fact that in isolation a regulator gene would not directly result in phenotypic resistance. The final analyses were performed using the R statistical language with the ggplot2 package v3.4.4 [35]. Abundant resistance classes were defined as those with >25,000 TMM-normalized counts.

### 2.7. Resistome and Mobilome Colocalization on Contigs

A subset of fecal samples was selected to achieve an even representation from differing ARG total abundances. Antimicrobial resistance gene TMM-normalized abundance was deemed as high (>30,000), medium (>10,000), or low (<10,000) at the ARG group level. Two of each abundance category were selected from each feedlot, resulting in a total of 24 fecal samples. Plasmids were constructed from assembled contigs using the MOB-recon tool from the MOB-suite package v3.0.0 [36] using default parameters. Using plasmid contigs as input, Staramr v0.7.1 [37] was used with default parameters to detect ARGs on plasmids. The output of both tools was subject to an in-house R script to correlate ARGs with their associated plasmids. Multi-drug resistance (MDR) plasmids were defined as those having ≥3 ARGs from different classes of antimicrobials. Chromosomal ARGs were assumed to be any contig that was not identified on a plasmid. Associations were constructed with SankeyMATIC (https://sankeymatic.com; accessed on 20 October 2023) and the circlize R package v0.4.15 [38].

### 2.8. Statistical Analysis

Comparisons between NAT and CONV beef production systems and normalized microbial taxa counts of interest were made using the Analysis of Compositions of Microbiomes with Bias Correction (ANCOM-BC) [39]. False discovery rates were mitigated through *p*-value corrections (q-value) using the Benjamini–Hochberg method [40] with the R package ANCOMBC v2.2.2. This same test was also applied to the normalized abundant (>25,000 TMM-normalized counts) ARG classes, mechanisms, and groups. For comparisons that were significant (q < 0.05), the fold change from CONV to NAT feedlots was calculated. Differences in Shannon diversity indices were determined with the Wilcoxon signed rank test (*p* < 0.05). Differences in Bray–Curtis distances were determined with PERMANOVA (*p* < 0.05).

## 3. Results

### 3.1. Microbiota Composition Differences between Conventional and Natural Systems

The phylum-level microbiota composition of the fecal samples was explored through normalized total abundance (Figure 1A). For fecal samples, CONV feedlots had higher (q < 0.05) abundances of Bacteroidetes, Euryarchaeota, Proteobacteria, and Spirochaetes. Fecal samples from CONV feedlots also had lower (q < 0.01) abundances of Actinobacteria and Tenericutes, while there was no difference in Cyanobacteria or Firmicutes. In catch basin water, no phyla differed in abundance (q > 0.05) between CONV and NAT feedlots (Figure 1B). Only fecal samples had increased alpha diversity from CONV to NAT feedlots (*p* < 0.0001; Appendix A) and a difference in beta diversity between feedlots (*p* < 0.001; Appendix A).

To quantify some of the differences in the microbiota between NAT and CONV feedlots, mean normalized abundances of taxa at classifications lower than phylum level were compared (Appendix A). Of the 17 most relatively abundant (>1%) archaeal and bacteria classes in fecal samples, 7 were significantly lower in CONV vs. NAT feedlots. The abundance of the classes Methanobacteria and Spirochaetia was lower by 1.7 and 1.5 fold, respectively, in NAT compared to CONV feedlots. For the catch basin water samples, classes did not differ between NAT and CONV feedlots.

The 20 most abundant genera in fecal and catch basin water samples are reported in Table 1. Two genera of note, *Methanobrevibacter* and *Treponema*, exhibited a 1.8- and 1.6-fold decrease (q < 0.01) in NAT vs. CONV production systems. A total of five genera (*Bacteroides*, *Chryseobacterium*, *Methanobrevibacter*, *Prevotella*, and *Treponema*; Figure 2 and Figure 3) were lower (q < 0.05) when NAT vs. CONV feedlot feces were compared, whereas seven genera (*Bacillus*, *Blautia*, *Clostridium*, *Eubacterium*, *Lactobacillus*, *Pseudomonas*, and *Streptococcus*; Figure 3) were higher in relative abundance (q < 0.01). Genera in catch basin water samples collected from CONV and NAT feedlots did not differ (q > 0.05). Five genera that were found to be abundant in both catch basin water and feces in both feedlot types were *Bacteroides*, *Clostridium*, *Flavobacterium*, *Prevotella*, and *Pseudomonas* (Table 1).

**Table 1 microorganisms-11-02982-t001:** Trimmed mean of m-value normalized mean relative abundance of the 20 most relatively abundant genera identified in fecal and catch basin water samples from conventional (CONV) and natural (NAT) feedlots of mean normalized abundance, interpreted significance of adjusted *p*-value (q > 0.05 = ns; q < 0.05 = *; q < 0.01 = **; q < 0.001 = ***; q < 0.0001 = ****), and fold change for significant differences.

Sample Type	Genus	CONV Mean Normalized Relative Abundance	NAT Mean Normalized Relative Abundance	Interpreted Significance	Fold Change
Fecal Composite	*Alistipes*	0.015	0.016	ns	-
	*Bacillus*	0.017	0.020	***	1.07
	*Bacteroides*	0.092	0.075	****	0.82
	*Blautia*	0.012	0.015	*	1.13
	*Butyrivibrio*	0.011	0.011	ns	-
	*Chryseobacterium*	0.014	0.011	****	0.80
	*Clostridium*	0.038	0.045	ns	-
	*Eubacterium*	0.013	0.016	ns	-
	*Faecalibacterium*	0.029	0.033	ns	-
	*Flavonifractor*	0.011	0.013	ns	-
	*Lachnoclostridium*	0.017	0.019	ns	-
	*Lactobacillus*	0.009	0.011	***	1.06
	*Methanobrevibacter*	0.015	0.006	****	0.55
	*Oscillibacter*	0.025	0.028	ns	-
	*Paenibacillus*	0.013	0.014	*	1.03
	*Prevotella*	0.071	0.051	*	0.77
	*Pseudomonas*	0.015	0.015	ns	-
	*Ruminococcus*	0.013	0.015	ns	-
	*Streptococcus*	0.009	0.011	**	1.04
	*Treponema*	0.028	0.012	**	0.63
Catch Basin Water	*Acidovorax*	0.014	0.017	ns	-
	*Aeromonas*	0.004	0.013	ns	-
	*Allochromatium*	0.018	0.003	ns	-
	*Arcobacter*	0.014	0.011	ns	-
	*Bacteroides*	0.004	0.013	ns	-
	*Bordetella*	0.012	0.009	ns	-
	*Brevundimonas*	0.011	0.005	ns	-
	*Burkholderia*	0.014	0.011	ns	-
	*Clostridium*	0.004	0.011	ns	-
	*Desulfomicrobium*	0.084	0.061	ns	-
	*Flavobacterium*	0.008	0.009	ns	-
	*Hydrogenophaga*	0.019	0.007	ns	-
	*Marichromatium*	0.014	0.002	ns	-
	*Methylomicrobium*	0.017	0.001	ns	-
	*Polynucleobacter*	0.010	0.067	ns	-
	*Prevotella*	0.001	0.012	ns	-
	*Pseudomonas*	0.058	0.053	ns	
	*Streptomyces*	0.023	0.026	ns	
	*Thauera*	0.026	0.012	ns	
	*Thiocystis*	0.020	0.003	ns	

**Figure 2 microorganisms-11-02982-f002:**
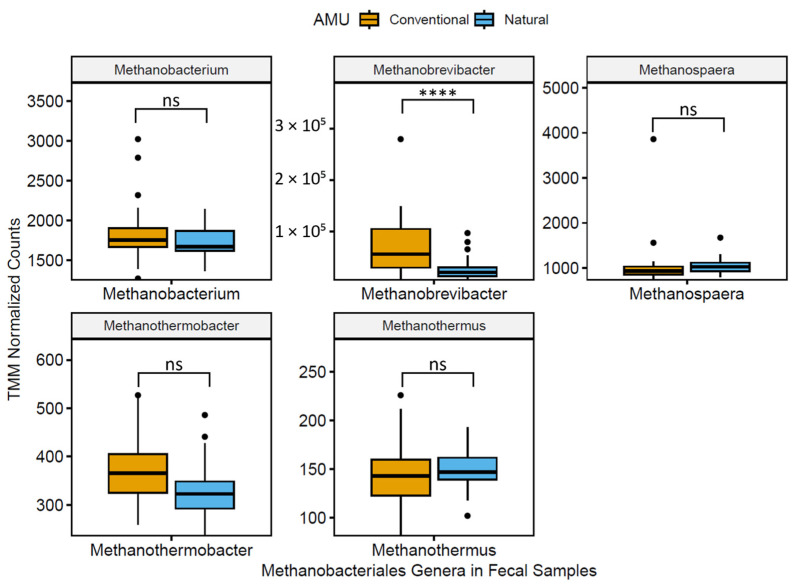
Boxplots of normalized TMM (trimmed mean m-value) abundance of prevalent (>1%) fecal genera within order Methanobacteriales (analysis of compositions of microbiomes with bias correction with adjusted *p*-value significance via Benjamini–Hochberg method; q > 0.05 = ns; q < 0.0001 = ****).

**Figure 3 microorganisms-11-02982-f003:**
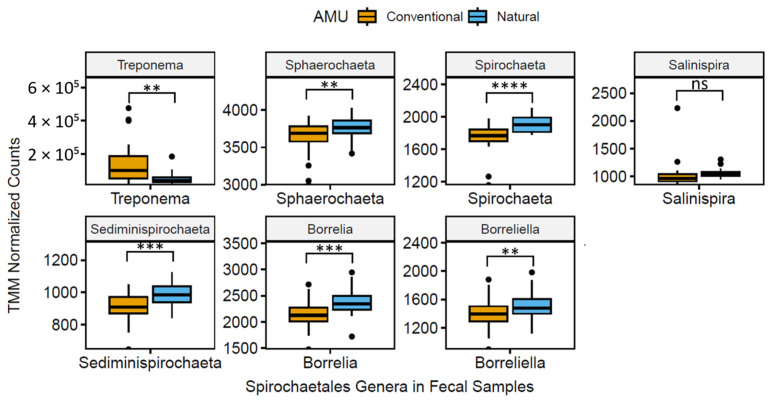
Boxplots of normalized TMM (trimmed mean m-value) abundance of prevalent (>1%) fecal genera within order Spirochaetales (analysis of compositions of microbiomes with bias correction with adjusted *p*-value significance via Benjamini–Hochberg method; q > 0.05 = ns; q < 0.01 = **; q < 0.001 = ***; q < 0.0001 = ****).

When determining the effect that diet may have had on the NAT microbiomes, only Spirochaetes increased (q < 0.0001) from backgrounding to finishing NAT cattle (Appendix A). When comparing the CONV vs. NAT cattle on finishing diets, many of the same differences were observed as in the overall CONV vs. NAT feedlots with the exception that there were no differences (q > 0.05) observed for Proteobacteria, Spirochaetes, and Tenericutes (Appendix A).

### 3.2. Resistome Differences between Conventional and Natural Systems

When all antimicrobial resistance classes (inclusive of biocides and metals) in fecal samples were analyzed, only class resistance types with >25,000 TMM-normalized counts were considered abundant, with the overall abundance of ARG classes’ presented in Appendix A. All samples showed similar resistance profiles, with genes conferring resistance to tetracyclines, MLS (macrolides, lincosamides, streptogramins), β-lactams, and aminoglycosides being most frequently identified (Figure 4A). Tetracyclines were the most abundant ARGs with TMM-normalized counts exceeding 60,000. The MLS ARGs were the next most abundant with over 25,000 TMM-normalized counts. Aminoglycosides and β-lactams did not differ, with <10,000 TMM-normalized counts. There were no significant differences among antimicrobial classes in fecal samples from CONV vs. NAT feedlots.

Catch basin water samples revealed many more prevalent ARG classes than feces with abundance > 25,000 TMM-normalized counts, including the presence of genes encoding for biocide and metal resistance (Figure 4B,C). Five drug resistance classes in catch basin samples were the same as those in fecal samples with the exception of the multi-drug resistance class in catch basin water (Figure 4B). There were also nine biocide and metal resistance classes in catch basin water, including arsenic, multi-biocide and metal, copper, multi-drug and biocide, iron, multi-biocide, multi-metal, peroxide, and tellurium resistance (Figure 4C). Again, similar to fecal samples, no differences (q > 0.05) were observed in most of the resistance classes associated with catch basin water, with tellurium resistance (q < 0.0001) being the exception.

The resistance groups from the most abundant classes were visualized as heatmaps with groupings according to the resistance mechanism for feces (Figure 5) and catch basin water (Figure 6). For both the CONV and NAT fecal samples, ARGs conferring resistance to tetracyclines were most abundant, with *tet*W, *tet*Q, *tet*O, *tet*44, *tet*40, and *tet*32 predominating. Aminoglycoside O-nucleotidyltransferases were the next most abundant ARGs, with *ant*(9) and *ant*(5) being the most common. There were a number of ARGs that were lower in NAT than in CONV feedlots that had an abundance of >10,000 TMM-normalized counts, including *mef*A (q < 0.001), *tet*40 (q < 0.001), *tet*O (q < 0.05), *tet*Q (q < 0.0001), and *tet*W (q < 0.05).

For catch basin water, the main variation in the resistance group was associated with individual samples as opposed to feedlot management type. One outlier of note was a catch basin water sample from NAT Feedlot B that had the same resistance profile pattern as a fecal sample. Most CONV and NAT catch basin samples also possessed genes encoding for biocide and metal resistance proteins (*ruv*B), copper resistance proteins (*actPC*), lincosamide nucleotidyltransferases (*lnu*C), and multi-metal resistance proteins (*zipB*, *recGM*).

When determining the effect that diet may have had on the resistome of NAT cattle, backgrounding vs. finishing NAT samples did not differ in ARG class abundance (q > 0.05; Appendix A). Likewise, for finishing diets, CONV and NAT cattle did not differ in ARG class abundance (q > 0.05; Appendix A).

### 3.3. Mobilome and Resistome Colocalization on Fecal Metagenomic Assemblies

Out of 24 fecal samples, 22 had at least one plasmid (Table 2). The two samples that lacked plasmids were from the high-level resistance category (>15,000 normalized counts). There were two instances of MDR, both occurring in CONV samples (Table 2). There were three cases of multiple aminoglycoside resistance genes within a single sample including at least two of *ant*(6)-*Ia*, *ant*(6)*-Ia*, or *aph*(3’)-*IIIa*.

There were four resistance profiles containing a single ARG within a plasmid (Figure 7A). The most prevalent AMR profile had genes conferring resistance to tetracyclines (36.8%) and aminoglycosides (18.4%). The next most prevalent resistance profile was a combination of aminoglycosides and tetracycline ARGs (15.8%). One of the instances of MDR was the presence of aminoglycoside, β-lactam, sulfonamide, and tetracycline ARGs on a single plasmid. The other instance was a plasmid harboring ARGs encoding for aminoglycoside, chloramphenicol, sulfonamide, and tetracycline resistance. The first MDR plasmid was predicted to be novel (i.e., no match in any database) and conjugative with a host range within the family *Enterobacteriaceae*. The second MDR plasmid was coded as AC935 and was predicted to be non-mobilizable with a host range within the genus *Acinetobacter*.

For chromosomal ARGs, there was an even distribution between CONV and NAT feedlots, whereas there were more plasmid-derived ARGs in CONV as compared to NAT feedlots (Figure 8). A larger proportion of aminoglycoside ARGs were found on plasmids as compared to tetracycline ARGs. Almost all MLS ARGs were chromosomally encoded. Both chloramphenicol and β-lactam ARGs were more likely to be found on the chromosome rather than on a plasmid.

For the ARG classes of interest, tetracyclines and aminoglycosides were more associated with CONV than NAT feedlots, but MLS did not differ between production systems (Figure 9). For these same ARG classes, all three were more often chromosome than plasmid associated. Of all ARG classes, tetracycline and aminoglycosides were most commonly plasmid associated. Chromosome-associated MLS ARGs were more prevalent than plasmid-associated ones in both CONV (20.3 fold) and NAT feedlots (41.0 fold). Tetracycline ARGs exhibited a similar pattern, but of a lower magnitude, with CONV being 5.6-fold greater and NAT 9.0-fold greater for chromosome-associated compared to plasmid-associated ARGs.

## 4. Discussion

The aims of this study were to observe the effect of antimicrobial use in beef cattle feedlots on the microbiome and resistome using metagenomics. Through the use of shotgun short-read (i.e., Illumina HiSeq2000) sequencing, trends and changes in microbiome and resistome composition were determined in fecal and catch basin water samples from both natural and conventional beef cattle feedlots. There is conflicting evidence on whether AMU impacts the phylum composition of cattle fecal microbiomes [16,41,42]. One study found an increased relative abundance of Bacteroidetes, Firmicutes, and Spirochaetes in the feces of cattle treated with antimicrobials [43]. Another found a similar trend, with the increased relative abundance of Proteobacteria and Firmicutes in CONV cattle fecal microbiomes, wherein they used a model that was able to account for only 0.6% of variance due to antimicrobial exposure [16]. This lack of model predictability, in addition to the many studies that failed to find large-scale differences in the microbiomes of cattle treated with or without antimicrobials [42,44,45], suggests that antimicrobials may not be responsible for major shifts in bacterial populations within the gastrointestinal tract of cattle. Instead, the variation that our study observed between feedlot systems might be influenced by other factors that differ between feedlot management practices.

Studies have found that grain-based diets are a driver for phylum-level microbiome instability [46] and that high-concentrate diets tend to decrease the overall relative abundance of most taxa [47]. These studies reflect the conditions of the CONV diets (80% barley grain, 15% barley silage, 5% supplement). Our study found that the finishing diets had higher relative abundances of Bacteroidetes and Euryarchaeota than backgrounding diets, while Corrêa et al. [47] found increases of only Firmicutes. This discrepancy might be explained by the high degree of microbiota instability (ability to maintain composition with perturbations) and variation attributed to high-grain diets [46].

Several classes were enriched in CONV fecal samples, including Bacteroidia, Epsilonproteobacteria, Flavobacteriia, Gammaproteobacteria, Methanobacteria, and Spirochaetia. These classes encompass both Gram-positive and Gram-negative bacteria along with archaea (Methanobacteria). The notion that macrolides select for certain bacterial taxa in bovine fecal samples is supported in the literature [48,49,50]. In particular, macrolide-resistant enterococci have been isolated from beef cattle feces in increasing proportions depending on the duration that tylosin is included in the diet [48,50]. Enterococci belong to the class Bacilli, contradicting our study as Bacilli were more abundant in NAT feedlots. Another study found that the relative abundance of Bacteroides increased with tylosin and monensin in feed [49]. There are few studies that have investigated the effect of AMU on the class taxonomic level, with most being restricted to the phylum level.

Few differences were found in the phyla, classes, orders, and genera between catch basin water samples obtained from NAT vs. CONV feedlots. This observation may be due to a dilution effect, as the members of the microbiota in catch basin water are not as directly influenced by diet as those that reside in feces. Instead, the higher proportions of Proteobacteria that thrive in catchment basins are likely to reduce differences between NAT and CONV feedlots.

The relative abundance of two genera, *Methanobrevibacter* and *Treponema*, was lower by >1.5 fold in NAT vs. CONV production systems. *Methanobrevibacter* spp. are methanogens that are found primarily in the gastrointestinal tract of animals such as termites, ruminants, and humans [51,52,53] and produce methane as a by-product of cellular respiration [54]. Monensin has been shown to target Gram-positive bacteria that produce hydrogen and formate, which are substrates for methanogenesis [55] It would have been predicted that the addition of monensin to the diets of cattle in CONV feedlots would result in a lower abundance of methanogens, such as *Methanobrevibacter* [56]. Instead, we found the opposite trend, with a 1.8-fold increase in the abundance of *Methanobrevibacter* in CONV vs. NAT feedlot types. This discrepancy highlights the impact of highly unstable microbiota composition in CONV AMU that may obscure any differences that the addition of monensin might induce. Additionally, *Treponema* spp. are members of the Spirochaetes that are typically considered to be commensals but can also be associated with bovine digital dermatitis [57]. Avoiding practices associated with an increased abundance of *Treponema* could theoretically reduce the case rate of bovine digital dermatitis and the attendant AMU. However, this study cannot conclusively identify the management practice(s) responsible for this proposed association.

As with the microbiome, in which the fecal and catch basin water samples were distinct, so were the resistomes. At the resistance class level, there were differences between the fecal and catch basin water samples. Tetracyclines and MLS ARGs dominated fecal samples, whereas, in the catch basin, all ARGs were present at similar levels. There were more ARGs in fecal samples than in catch basin water, as expected, due to the higher microbial density in fecal samples. A previous study found that fecal composite and catch basin water samples shared 83% of ARG groups and were separated into distinct clusters following nonmetric multidimensional scaling analysis [17]. The single catch basin water resistome profile that resembled a fecal composite resistome profile should be treated as a source of “fecal contamination”, and likely arose as a result of a significant run-off event. This is encouraging because it indicates that the resistomes associated with bacteria in fecal composite samples are likely contained within the catchment basin.

Of the ARGs examined, the mechanisms that had significantly lower abundance in the NAT as compared to the CONV fecal samples were lincosamide O-nucleotidyltransferases, MLS resistance MFS efflux pumps, and tetracycline ribosomal protection proteins. While significant, the low fold difference of ~1 indicated that there was essentially no difference between feedlots as a fold difference of 1 has a ratio of 1:1 (i.e., 100% similarity). As phylum composition in the fecal microbiome may be attributed to diet, so too might the resistome. This conclusion is supported by evidence that certain ARGs tend to be found in a limited number of taxa, as even low-diversity communities can have high levels of ARGs [58]. The transition of weaned calves onto a solid diet has been associated with increased ARG prevalence, with the exception of tetracycline and MLS ARGs, which were shown to increase [59]. This suggests that starch digestion is associated with bacteria with a higher prevalence of MLS and tetracycline ARGs. A recent study that sampled NAT beef cattle fed a diet similar to ours [60] found that expressed ARGs in the rumen at slaughter were not associated with AMU, but rather with bacterial metabolic pathways.

Intrinsic ARGs are more likely to be contained within a chromosome than on a plasmid due to ‘intrinsic’ meaning belonging to all members of a species (i.e., belonging to the core genome) [61]. In a similar case, but not to the same extent, acquired ARGs will more likely be found on mobilizable MGEs rather than be associated with the chromosome, such as ICE [8,62]. Given this trend, it could be inferred that MLS resistance is more conserved than tetracycline resistance since there were more tetracycline ARGs found on plasmids than MLS ARGs [63]. As tetracyclines and macrolides have both been highly used historically in beef cattle, one might predict that they should both be mainly associated with the chromosome. It is unclear why this trend was not apparent for both ARG classes, but it may have to do with co-selection. If there was a highly conserved gene (e.g., heavy metal resistance gene) in proximity to the chromosomal MLS ARG locus, then the MLS ARG could also be highly conserved [64]. While this may explain some differences, it is difficult to validate through metagenomics. Nevertheless, these data suggest that tetracycline ARGs are more mobile than MLS ARGs and that tetracycline ARGs are retained even in the absence of tetracycline use. This may reflect the generational selective pressures that tetracyclines have presented to the beef cattle resistome or the use of tetracyclines in backgrounding feedlots or cow–calf systems.

The main limitation of this study is that the contig coverage of the entire metagenome will never be as comprehensive as that of the read coverage. This means that the assembly step that is required to enable short-read data to investigate colocalization and the overall abundance of elements of interest will always lack precision. The employment of a hybrid assembly approach using short and long-read sequences would be one approach to improve the ability to define the colocalization of ARGs with specific MGE.

## 5. Conclusions

When comparing AMU practices, certain genera (*Treponema* and *Methanobrevibacter*) were enriched in fecal samples from cattle at feedlots that used antimicrobials. Few significant differences were observed in the microbiomes of catch basin samples from NAT vs. CONV feedlots. Contrary to our initial hypothesis, there were also no significant differences in the fecal resistome of cattle between the two feedlot management systems. It seems that generationally selected resistomes through decades of AMU persist even after antimicrobials are not used within the beef production system. The minute differences observed at the ARG group level may be explained by other differences in AMU and diet between CONV and NAT feedlots. More comprehensive investigations into the relationship between livestock diet and AMR should be conducted. Tetracycline selective pressure on the beef cattle industry is strongly established based on the historical use of this antimicrobial and the high abundance of tetracycline resistance genes present in the fecal microbiome. Our study suggests that short-term elimination of AMU is unlikely to substantially reduce the prevalence of ARGs in feedlot environments.

## Figures and Tables

**Figure 1 microorganisms-11-02982-f001:**
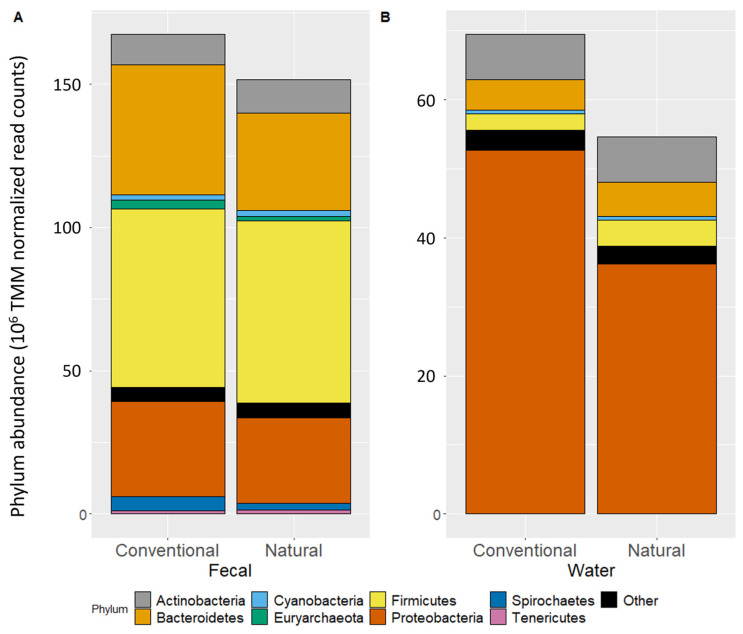
The abundance of phyla (>1%) in fecal (**A**) and catch basin water (**B**) samples normalized with the TMM (trimmed mean of m-values) method across conventional and natural feedlots.

**Figure 4 microorganisms-11-02982-f004:**
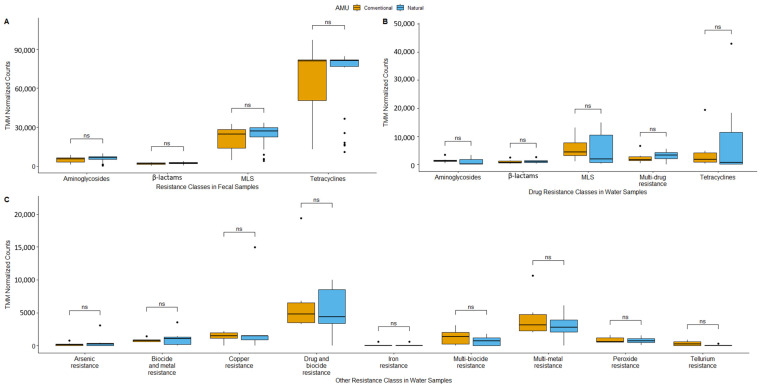
Boxplots of trimmed mean of m-value (TMM)-normalized gene abundance of resistance for abundant resistance classes (>25,000 normalized counts) in fecal samples (**A**); antimicrobial drug resistance classes in catch basin water samples (**B**); and biocide and metal resistance classes in catch basin water samples (**C**) (MLS = macrolide, lincosamide, streptogramin; analysis of the composition of microbiomes with bias correction and adjusted *p*-value significance used the Benjamini–Hochberg method; q > 0.05 = ns).

**Figure 5 microorganisms-11-02982-f005:**
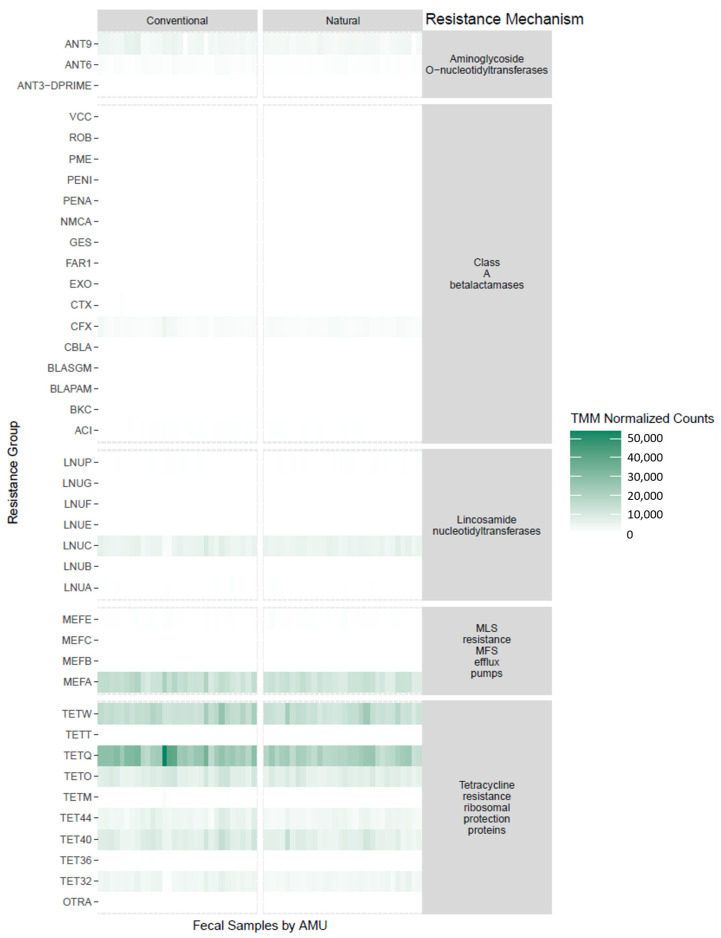
Heatmap of antimicrobial resistance groups from fecal samples stratified into resistance mechanism comparing conventional and natural feedlots.

**Figure 6 microorganisms-11-02982-f006:**
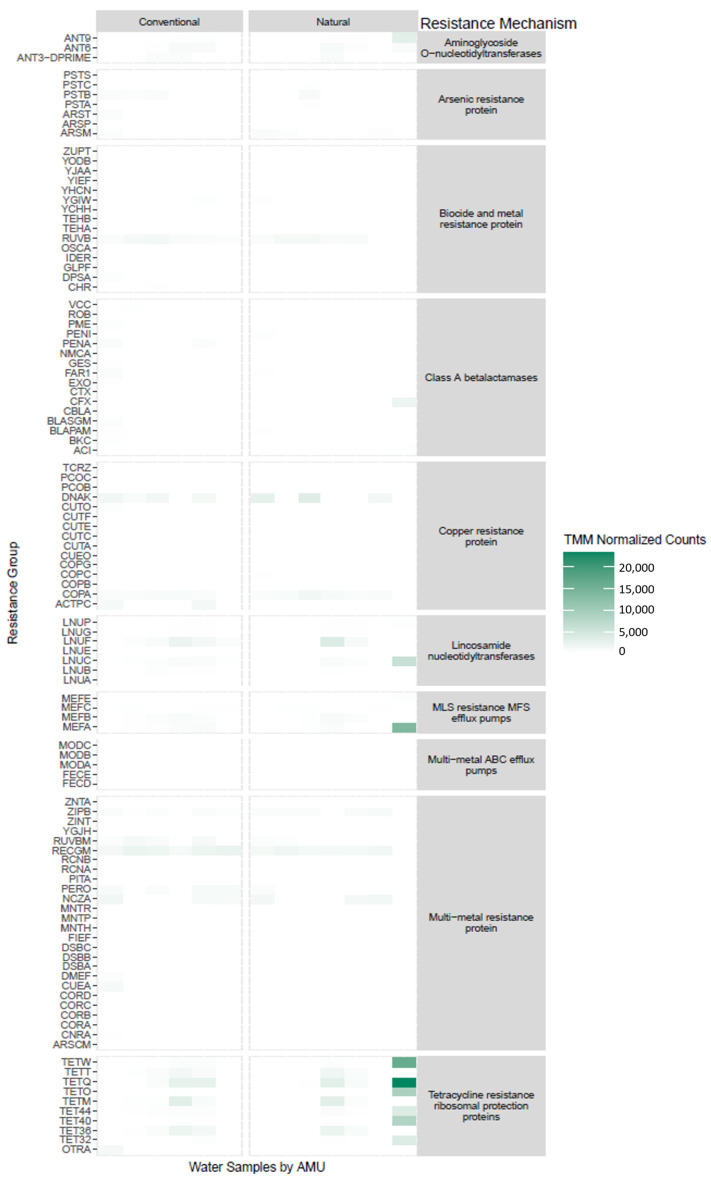
Heatmap of antimicrobial resistance groups in catch basin water samples stratified into resistance mechanism comparing conventional and natural feedlots.

**Figure 7 microorganisms-11-02982-f007:**
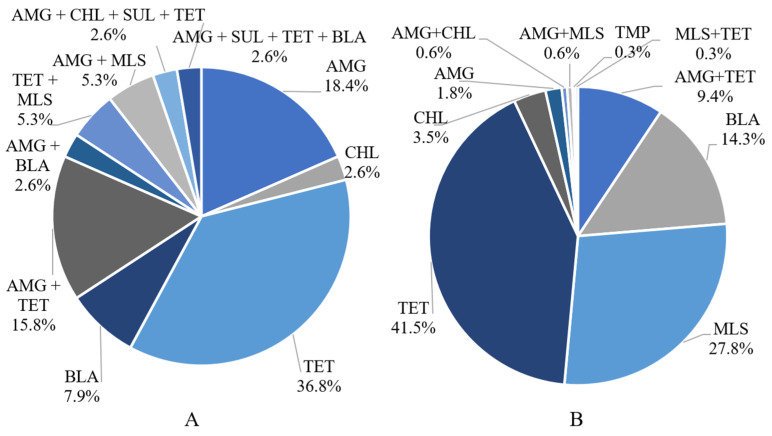
Fecal resistance profiles of ARG-carrying contigs from (**A**) 102 plasmids (*n* = 24 samples) and (**B**) 342 chromosomes (*n* = 60 samples). AMG = aminoglycoside; BLA = β-lactam; CHL = chloramphenicol; MLS = macrolide, lincosamide, and streptogramin; SUL = sulfonamide; TET = tetracycline; TMP = trimethoprim. With regard to chromosomal ARGs, the most abundant resistance class was tetracycline (41.5%) followed by MLS (28.7%), β-lactam (14.3%), chloramphenicol (3.5%), aminoglycoside (1.8%), and trimethoprim (0.3%; (**A**)). In most cases, no two ARGs belonging to the same class were associated with a contig, with the exception of two instances of two aminoglycoside ARGs, *ant*(6)-*Ia* and *aph*(3’)-*III*, and one instance of two tetracycline ARGs, *tet*(40) and *tet*(O).

**Figure 8 microorganisms-11-02982-f008:**
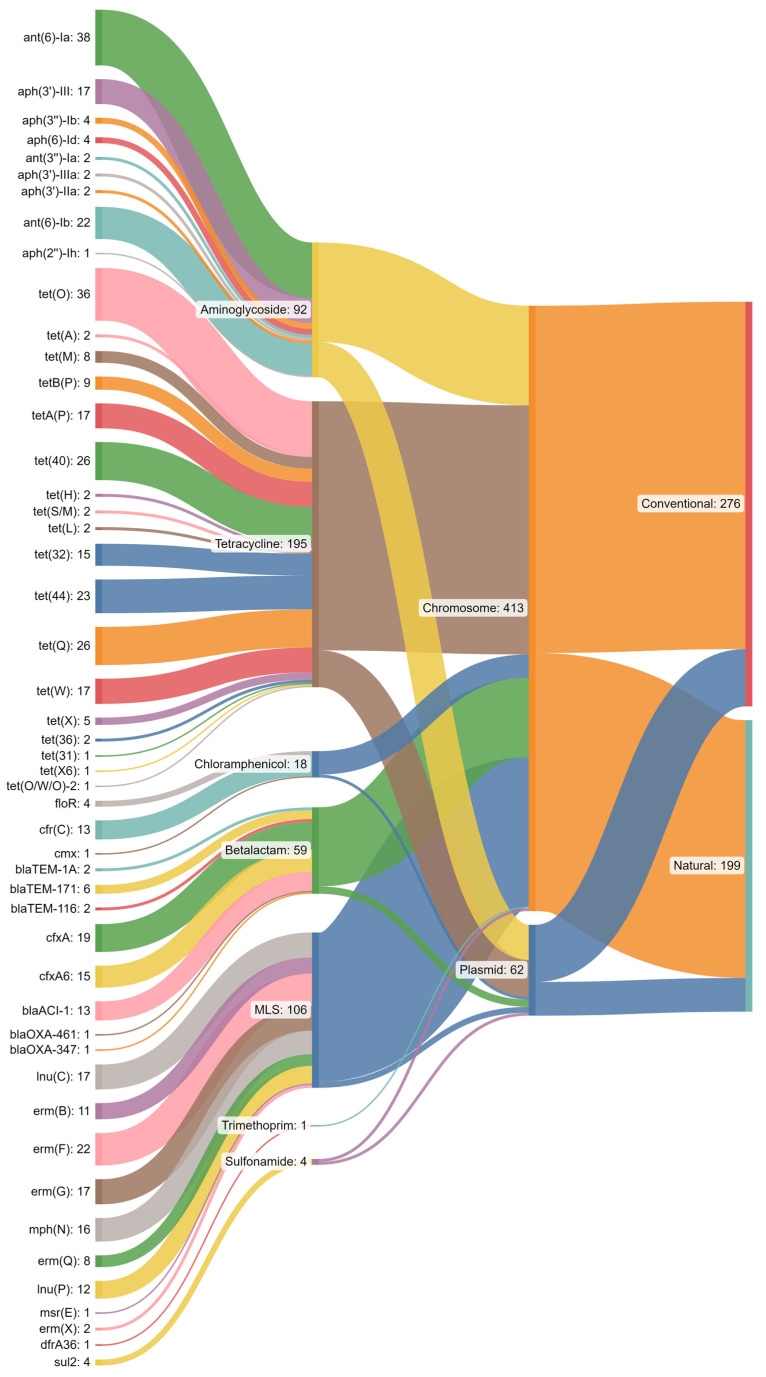
Sankey diagram depicting contigs with antimicrobial resistance genes (ARGs) and antimicrobial classes associated with their respective contig localizations and feedlot type.

**Figure 9 microorganisms-11-02982-f009:**
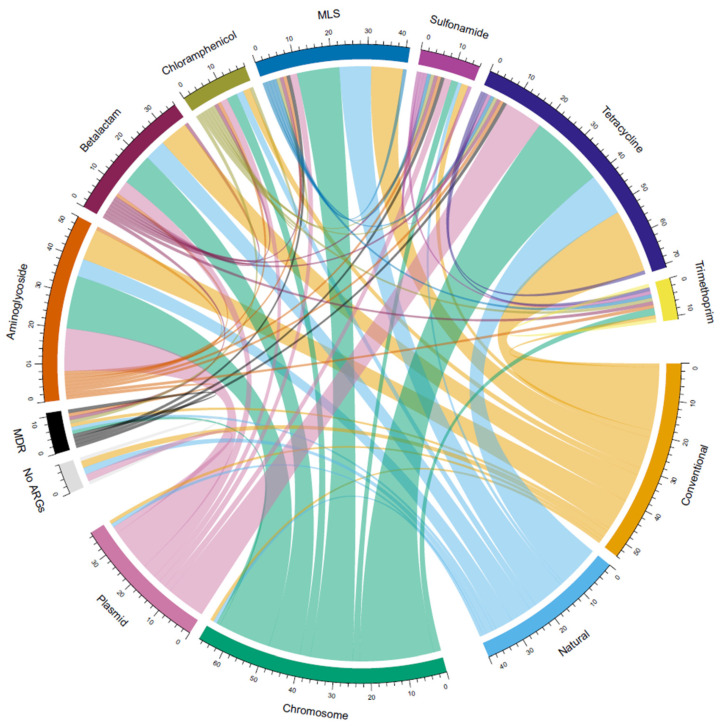
Chord diagram of associations between antimicrobial resistant gene classes in conventional vs. natural feedlots, and association with chromosomes or plasmids. Associations within the same antimicrobial resistance gene (ARG) reflect that there are multiple ARGs on the same contig that confer resistance to a single antimicrobial class. (MDR = multi-drug resistance from a single antimicrobial resistance gene).

**Table 2 microorganisms-11-02982-t002:** Summary of colocalized antimicrobial resistance gene-carrying plasmids from assembled contigs.

Level of Resistance ^1^	AMU	Sample ID	No. of Plasmids with ARGs	No. of MDR ^2^-Carrying Plasmids
Low	CONV	Con-Sum-A-05Sep17-MG44	1	-
		Con-Sum-A-26Apr17-MG7	3	1
		Con-Sum-D-26Jun18-MG19	1	-
		Con-Sum-D-26Jun18-MG20	1	-
	NAT	Nat-Sum-B-26Apr17-MG6	2	-
		Nat-Win-B-13Mar18-MG14	3	-
		Nat-Win-C-29Mar17-MG58	1	-
		Nat-Win-C-31Jan17-M7	1	-
Medium	CONV	Con-Sum-D-1Aug17-MG28	2	-
		Con-Win-A-11Dec17-MG55	1	-
		Con-Win-A-13Mar18-MG15	2	-
		Con-Win-D-2Feb17-M10	3	-
	NAT	Nat-Sum-B-26Apr17-MG5	1	-
		Nat-Sum-C-10Apr18-MG9	1	-
		Nat-Sum-C-26Jun18-MG17	1	-
		Nat-Win-B-25Oct16-M1	2	-
High	CONV	Con-Win-A-1Mar17-MG3	2	-
		Con-Sun-A-25Jul18-MG23	0	
		Con-Win-D-1Feb17-M9	4	1
		Con-Win-D-29Mar17-MG59	2	-
	NAT	Nat-Sum-B-28Jun17-M18	2	-
		Nat-Sum-B-29May18-MG45	0	-
		Nat-Sum-C-1Aug17-MG25	1	-
		Nat-Win-C-31Jan17-MG50	1	-

^1^ Levels of resistance as TMM-normalized reads: high (>15,000), medium (>5000), and low (<5000). ^2^ MDR: multi-drug resistance defined at resistance to ≥3 antimicrobial classes.

## Data Availability

All Illumina sequence read data from the current study have been deposited to the NCBI database as Short Read Archive (SRA) under BioProject ID PRJNA420682. These data are publicly available at https://www.ncbi.nih.gov/bioproject/ (accessed on 11 December 2023).

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
