# Peer review of "Effect of Antimicrobial Use in Conventional Versus Natural Cattle Feedlots on the Microbiome and Resistome"

_microorganisms, 2023, doi:10.3390/microorganisms11122982_

Round 1
Reviewer 1 Report
Comments and Suggestions for Authors
This study compared the prevalence and abundance of ARGs and the coexistence of ARGs and MGE in conventional and natural beef cattle feedlots in Alberta. This work shows that if antibiotics stop in the natural production system, after decades of generation selection resistance still exists, which is worth pondering. The samples are representative and the number of samples is sufficient for the analysis, which is a comprehensive and difficult task. In addition, the analysis data of this work is sufficient and abundant. The idea of this work is innovative and reasonably analyzed. Some details should be considered and addressed before the manuscript is accepted. Details are as follows:
1. Please evenly distribute the text between the margins in the text of the manuscript.
2. Table 1. Please standardize the use of tables and neatly simplify the arrangement otherwise it seems to be confusing.
3. Line 71. Please delete the extra full point.
4. Figure 4, 5, 6, 8. The description of the picture is unclear and small, please deepen the font color.
5. Line 418-421. This sentence is difficult to understand. Some studies have shown the results of speculation but the author has not come to the opposite conclusion. Please re-describe this sentence.
6. Table S1. Please standardize the use of tables and neatly simplify the arrangement.
Comments on the Quality of English LanguageMinor editing of English language required.
Reviewer 2 Report
Comments and Suggestions for Authors
Effect of Antimicrobial Use in Conventional Versus Natural Cattle Feedlots on the Microbiome and Resistome
The paper, titled “Effect of Antimicrobial Use in Conventional Versus Natural Cattle Feedlots on the Microbiome and Resistome“ aims to investigate the impact of Conventional (CONV-raised with antimicrobials and hormones) and Natural (NAT-raised without antimicrobials and hormones) beef cattle feedlots in Alberta, Canada.
This study addresses an increasingly important issue of current interest, covering several salient aspects of the subject and making a fundamental contribution to scientific research in this area. In addition to the antimicrobial resistance analysis performed on metagenomic DNA extracts derived from fecal samples, data were also processed to assess microbial taxonomy, resistome, microbial diversity, and the correlation between antimicrobial resistance genes and plasmids.
I found the subject matter of the article fascinating and read the manuscript with great interest. The topic discussed is highly relevant and intriguing, aligning well with the journal's scope. However, there are several areas that could benefit from improvement.
Main question
The main question addressed by the research is to investigate the phenomenon of antimicrobial resistance (AMR) in conventional and natural beef cattle feedlots, predicting that the antimicrobial use (AMU) in the conventional feedlots results in increased prevalence and abundance of antimicrobial resistant genes (ARGs) and ARG colocalized with mobile genetic elements (MGEs) compared with natural cattle farms.
Simple summary
I suggest writing the simple summary. According to the author's guidelines, this section should summarize and contextualize your paper within the existing literature in your field. It should be written without technical language or nonstandard acronyms, with the goal of conveying the meaning and importance of this research to non-experts.
Abstract
The abstract is correlated with the manuscript content, but I recommend rewriting the text including more results and the significance of the obtained data.
Keywords
To enhance the research's appeal, I suggest revising the keywords, avoiding the inclusion of terms that are already present in the article title.
Introduction
I recommend providing a more detailed overview of the study's context, with an emphasis on incorporating references from existing literature. To enhance the introduction, consider beginning with a broader discussion on the challenges faced by the modern beef industry, particularly in the context of antimicrobial use within intensive farming systems. Support your points with relevant references to establish a comprehensive foundation for the study. As example: 10.1016/j.rvsc.2023.03.008.
In addition to the methods used and the expected results, which are already in the text at the end of the introductory section, I recommend highlighting more explicitly the goals set to be achieved, in order to make them clearer for readers.
Materials and Methods
The materials and methods section has some shortcomings that impede the research's reproducibility. I recommend rewriting the portion on experimental diets for better comprehension. Additionally, more details about the animals, such as sex, average weight, age, body condition scores, etc., should be included.
To enhance the transparency and replicability of your research, I kindly suggest that you include a section detailing the chemical composition of the diets.
Results
Table 2 (line 351-page 13) is not mentioned in the text.
Discussion
Starting the discussion section by reiterating the aim of the study can provide clarity and context for readers.
I kindly suggest expanding the discussion section of your paper to include practical applications and a thorough exploration of the study's limitations. This addition will enhance the overall value of your research and provide a more comprehensive understanding of its implications.
It would also be valuable to set up the discussion in divided sections that take up the different aspects addressed in the results, so as to make it easier to read and find the information in the text.
Conclusion
I kindly suggest expanding the conclusions section of your paper to provide a more detailed and comprehensive report of the main findings. This will help readers better understand the significance of your research.
References
All the references are appropriate and included in the main text.
Editing
There are some editing issues. It's recommended to thoroughly review the document for such problems.
Round 2
Reviewer 2 Report
Comments and Suggestions for Authors
well done!